# Midwives' survey of their weight management practice before and after the GLOWING guideline implementation intervention: A pilot cluster randomised controlled trial

**Nicola Heslehurst** *, **Catherine McParlin, Falko F. Sniehotta, Judith Rankin, Elaine McColl**

Population Health Sciences Institute, Newcastle University, Newcastle upon Tyne, Tyne and Wear, United Kingdom

* nicola.heslehurst@ncl.ac.uk

## Abstract

### Background

Maternal weight management is a priority due to pregnancy risks for women and babies. Interventions significantly improve maternal diet, physical activity, weight, and pregnancy outcomes. There are complex barriers to midwives' implementation of guidelines; low self-efficacy is a core implementation barrier. The GLOWING intervention uses social cognitive theory (SCT) to address evidence-based barriers to practice. The intervention aimed to support midwives' implementation of guidelines.

### Methods

An external rehearsal pilot cluster randomised controlled trial in four NHS Trusts (clusters) in England, UK. Clusters were randomised to intervention (where all eligible midwives received the intervention) or control (no intervention delivered) arms. A random sample of 100 midwives were invited to complete questionnaires pre- and post-intervention. UK guideline recommendations relating to midwives' practice were categorised into: 1) communication-related behaviours (including weight- and risk-communication), and 2) support/intervention-related behaviours (including diet/nutrition, physical activity, weight management, referrals/signposting). Questionnaires were developed using SCT constructs (self-efficacy, outcome expectancies, intentions, behaviours) and 7-point Likert scale, converted to a 0–100 scale. Higher scores were more positive. Descriptive statistics compared intervention and control arms, pre- and post-intervention.

### Results

Seventy-four midwives consented and 68 returned questionnaires. Pre-intervention, self-efficacy for support/intervention-related behaviours scored lowest. In controls, there was limited difference between the pre- and post-intervention scores. Post-intervention, mean (SD) scores were consistently higher among intervention midwives than controls,

could potentially be identifiable from the data. Requests to access the data must be made to the approving Ethics Committee (Proportionate Review Sub-committee of the Yorkshire & The Humber— South Yorkshire Research Ethics Committee (ref: 15/YH/0565, 16/12/2015), contact via southyorks. rec@hra.nhs.uk) for researchers who meet the criteria for access to confidential data.

**Funding:** Award received by NH and EM. National Institute for Health Research Postdoctoral Research Fellowship (Heslehurst, PDF-2011-04-034). https://www.nihr.ac.uk/researchers/apply-for-funding/how-to-apply-for-career-development-support/apply-for-an-award The funders had no role in study design, data collection and analysis, decision to publish, or preparation of the manuscript.

**Competing interests:** The authors have declared that no competing interests exist.

particularly for support/intervention self-efficacy (71.4 (17.1) vs. 58.4 (20.1)). Mean (SD) self-efficacy was higher post-intervention than pre-intervention for all outcomes among intervention midwives, and consistently higher than controls. Mean differences pre- and post-intervention were greatest for support/intervention self-efficacy (17.92, 95% CI 7.78–28.07) and intentions (12.68, 95% CI 2.76–22.59). Self-efficacy was particularly increased for diet/nutrition and physical activity (MD 24.77, 95% CI 14.09–35.44) and weight management (18.88, 95% CI 7.88–29.88) behaviours, which showed the largest increase in scores.

## Conclusions

This study supports the theoretical models used to develop GLOWING, where low self-efficacy was a core implementation barrier. Results suggest that GLOWING successfully targets self-efficacy, potentially with a positive impact on guideline implementation. A definitive trial is required to determine effectiveness.

## Trial registration

ISRCTN46869894, retrospectively registered 25/05/2016, http://isrctn.com/ISRCTN46869894.

## Introduction

More than half of women enter pregnancy with a body mass index (BMI) in the overweight (BMI$\geq$25kg/m$^2$) or obese range (BMI$\geq$30kg/m$^2$) in the UK, with one in five having obesity [1]. Supporting women with weight management during pregnancy is a public health priority due to the inequalities associated with maternal obesity, including significantly higher prevalence among women living in deprived locations and among Black and South Asian ethnic groups [2, 3]. Pregnancy is also a critical period for the development, or worsening, of obesity due to excessive gestational weight gain (GWG) and postnatal weight retention [4], and for intergenerational obesity development [5]. Addressing weight management during pregnancy is also a clinical priority for maternity services due to the obesity-related risks to women and their babies, including gestational diabetes, preeclampsia, and maternal and offspring mortality [6–8].

Weight management interventions delivered during pregnancy can significantly improve women's diet and physical activity (PA) behaviours, reduce GWG and postnatal weight retention, and reduce the risk of developing some adverse pregnancy outcomes [9–15]. The existing evidence-base identifies the importance of frequent and personal interactions with health professionals during the delivery of interventions [16, 17]. The absence of these interactions with health professionals is a barrier to intervention success. This highlights the need to embed weight management support into routine maternity care [18].

In the UK, national guidelines for weight management during pregnancy include recommendations relevant to health professionals' routine practice [19]. These broadly relate to health professionals' communication behaviours and the provision of support and intervention, including discussing women's weight and obesity-related risks, providing support for women's diet, PA and weight management and referring or signposting to additional support services and information sources (S1 Fig). However, there are complex barriers to health professionals' implementation of these guideline recommendations [20]. For example, health

professionals reported a lack of confidence in their knowledge of weight management, weight-related communication skills and beliefs that women would respond negatively to discussions about their weight, which would impact on their relationship [20]. Building health professional capacity to address maternal obesity and improve weight management support in pregnancy is an established need [21], with recommended areas of development relating to having knowledge and skills to provide weight management support, sensitive communication techniques, and knowledge of local services [19, 22].

The GestationaL Obesity Weight management: Implementation of National Guidelines (GLOWING) intervention was developed using social cognitive theory (SCT) to address the evidence-based barriers to the implementation of guidelines regarding weight management in pregnancy [20], with the aim of supporting midwives' implementation into routine practice [23]. SCT is based on the principles that the person, environment, and behaviour all interact and influence one another and that behaviours are directly related to an individual's behavioural goals [24]. In the context of this study, the theoretical construct at the core of SCT is midwives' self-efficacy; additional constructs include outcome expectancies and goals/intentions. The GLOWING intervention was delivered as a rehearsal (external) pilot trial with integrated process evaluation, therefore delivered as per a definitive trial but on a smaller scale [27]. The trial protocol was published [23] and registered on the ISRCTN (ISRCTN46869894). This paper uses the Standards for Reporting Implementation Studies (StaRI) Statement [25] (S1 Table).

Materials and methods

The aim of the GLOWING intervention was to support midwives' implementation of UK guidelines for weight management during pregnancy into routine practice. The aim of the implementation of these guidelines is to improve pregnant women's diet and physical activity behaviours and gestational weight gain. This paper reports the descriptive analysis of the midwives' self-reported SCT constructs related to implementation of the UK guideline recommendations.

## Design, setting and participants

The details of the GLOWING intervention methods have been described elsewhere [23, 26]. The pilot trial was an external rehearsal multi-centre parallel group cluster randomised clinical trial (RCT) comparing the delivery of the SCT-based behaviour change intervention for midwives versus usual practice. The clusters were four NHS Trusts which provide maternity care in the North East of England, UK; a region with high levels of deprivation (Fig 1). Two large and two small Trusts were included, based on average number of deliveries per year, and randomisation was stratified based on size of the Trust. The participants were community midwives and hospital-based midwives with a specific role in obesity or weight management. We aimed to deliver the GLOWING intervention to all eligible midwives in the intervention arm and, following guidelines on sample size requirements for pilot studies [27], to recruit a random sample of 30 midwives per intervention arm to provide outcome data in the form of questionnaires pre- and post-intervention.

## Intervention development and delivery

We followed a four-step approach for developing theory-informed implementation interventions [28], described elsewhere [23]. Part of this process involved systematically mapping the evidence-based barriers and facilitators to practice [20] to SCT models for both behaviour categories: communication-related UK guideline recommended behaviours (Fig 2) and support/intervention-related behaviours (Fig 3). The majority of evidence-based barriers identified

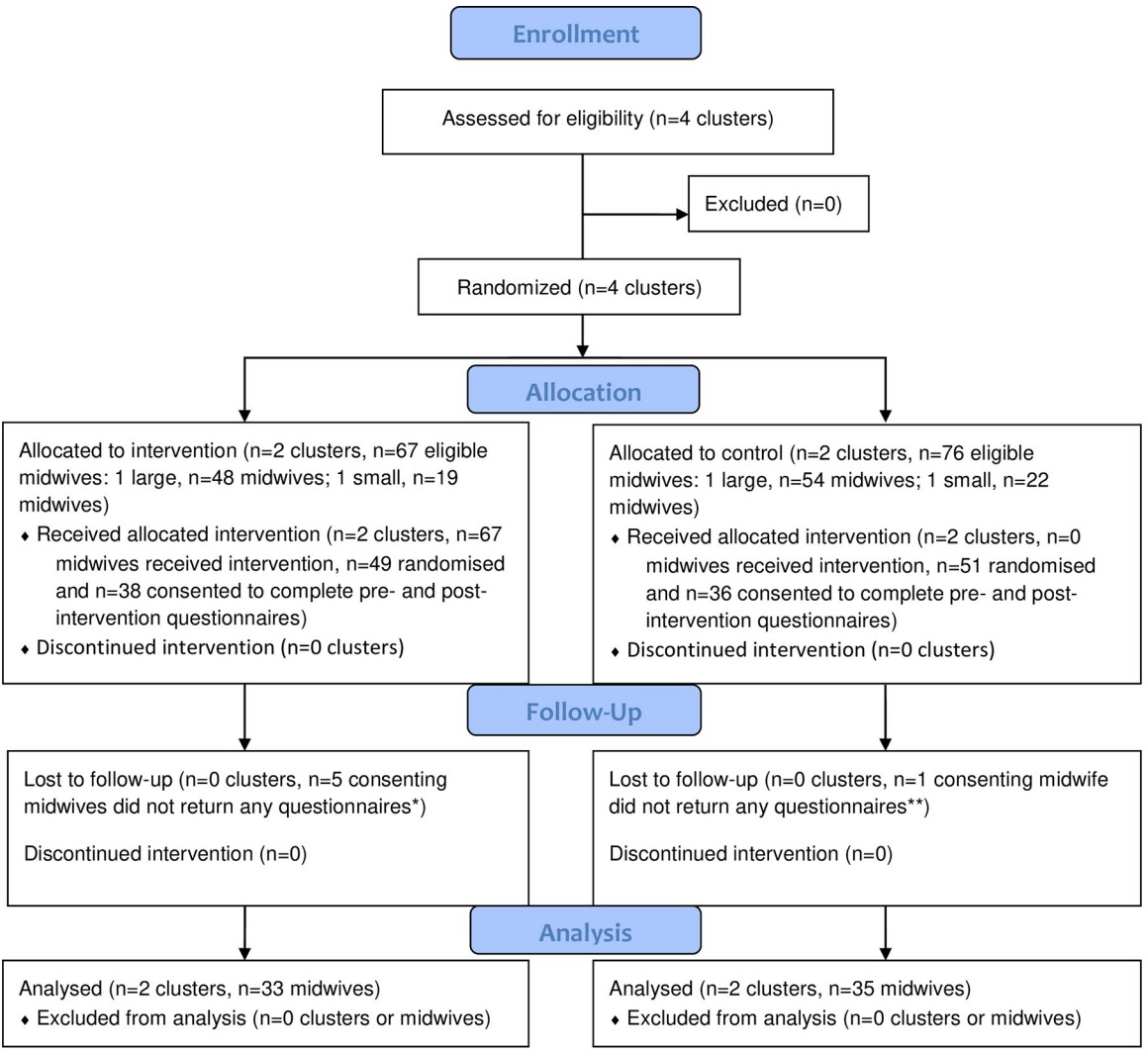

**Fig 1. CONSORT 2010 flow diagram for cluster trials.** * Intervention arm: n = 33 midwives returned ≥1 questionnaire: n = 32 pre-intervention, n = 15 post-intervention, n = 14 both ** Control arm: n = 35 midwives returned ≥1 questionnaire: n = 34 pre-intervention, n = 34 post-intervention, n = 33 both.

were for the support/intervention-related behaviours. Both SCT models included self-efficacy at their core as the primary barrier to guideline implementation. Low self-efficacy was related to health professionals' lack of confidence in their weight communication skills (Fig 2) and in their knowledge and skills to support women with behaviour change and weight management (Fig 3). Key determinants of communication-related self-efficacy were related to outcome expectancies, especially beliefs that women would react negatively to discussions about their weight status, and regarding obesity-related risks in pregnancy (Fig 2). Health professionals were concerned that this communication would impact on their relationship with the women,

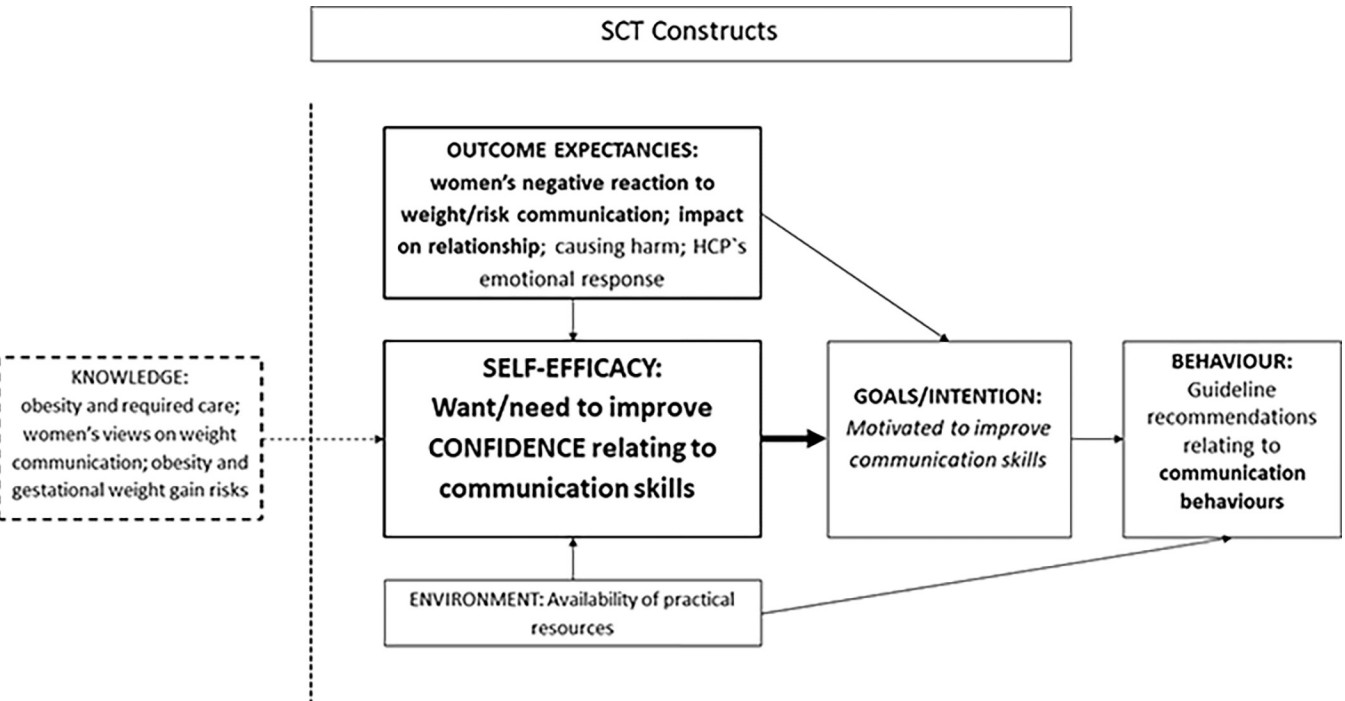

**Fig 2. SCT theoretical model for implementation of guideline recommendations for weight management in pregnancy: Communication-related behaviours.** SCT = social cognitive theory. Self-efficacy, outcome expectancies, goals/intentions and environment are SCT constructs relating to behaviours. Knowledge was identified as an important additional construct in the evidence base having an influence on self-efficacy. Bold and larger font size represent the key barriers to practice. Italics font represents facilitators to practice.

which was important to develop and maintain over the course of pregnancy. For the support/intervention-related behaviours, key determinants of self-efficacy were a lack of evidence-based weight management knowledge and expertise, at the level relevant to their professional role (Fig 3). Additionally, within this model, a lack of evidence-based knowledge of the determinants of obesity and weight gain was also related to negative attitudes towards people living with obesity held by health professionals. These influenced health professionals' outcome expectancies on the effectiveness of, and women's motivation and engagement with, weight management interventions in pregnancy.

There was consistency in the facilitators in both models relating to goals/intention, whereby health professionals were motivated to improve their communication and weight management skills (Figs 2 and 3), and to intervene to support women with weight management as they perceived this to be important and a priority area for practice (Fig 3). There were additional facilitators specific to support/intervention-related behaviours, as health professionals had some confidence in providing general healthy behaviour advice and experience of behaviour change skills in practice (Fig 3), which the intervention could build upon in the context of obesity and weight management.

The GLOWING intervention (content described in [20]) was developed to address the evidence-based barriers incorporated in these models, and to build on the facilitators, with particular emphasis on improving midwives' confidence (self-efficacy) in both behaviour categories. The intervention was delivered as an intensive face-to-face training session for small groups of midwives, plus the provision of information resources to share with pregnant women during routine practice.

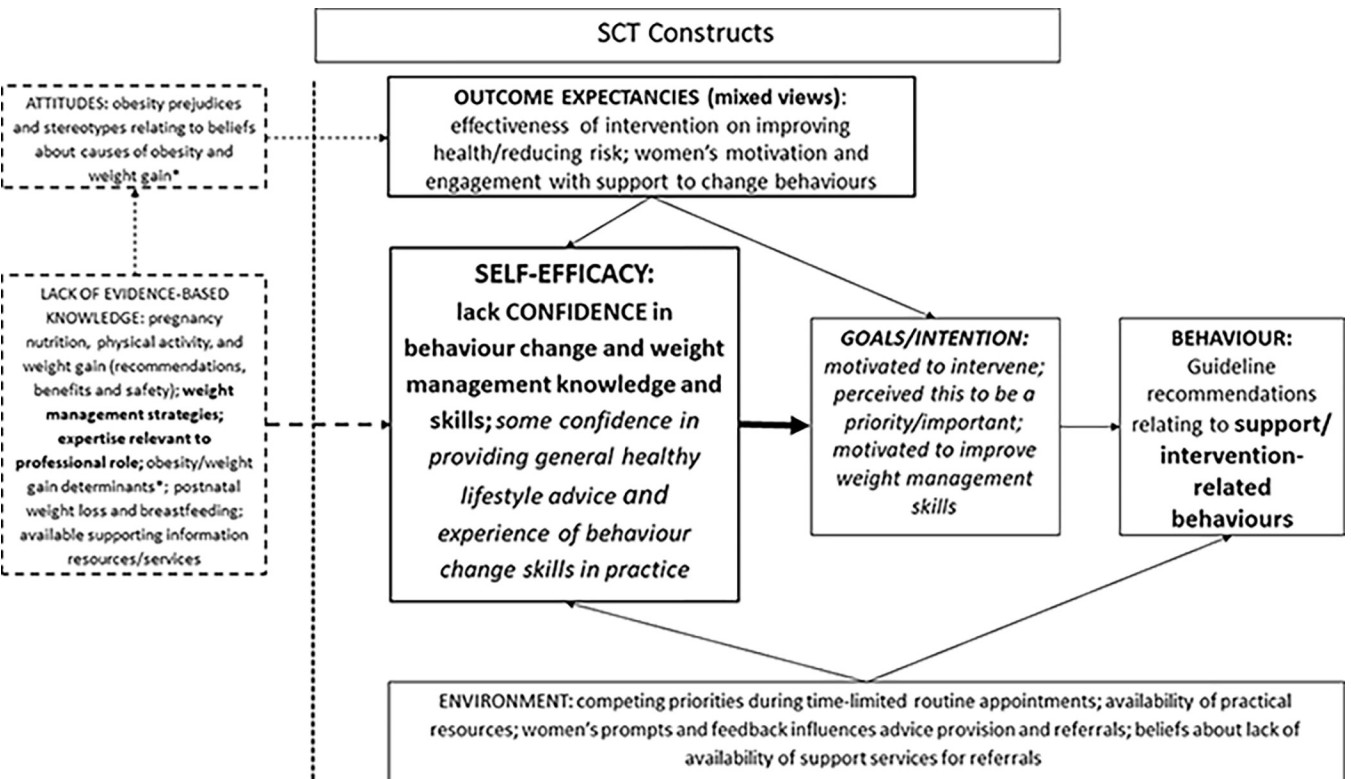

**Fig 3. SCT theoretical model for implementation of guideline recommendations for weight management in pregnancy: Support/intervention-related behaviours.** SCT = social cognitive theory. Self-efficacy, outcome expectancies, goals/intentions and environment are SCT constructs relating to behaviours. Knowledge and attitudes were identified as an important additional construct in the evidence base having an influence on self-efficacy and outcome expectancies. Bold and larger font size represent the key barriers to practice. Italics font represents facilitators to practice.

## Data collection

Data reported in this paper are from the questionnaires that midwives completed pre- and post-intervention. There were no existing validated questionnaires that could be used to measure the SCT constructs as the questions had to be tailored to the guideline specific behaviours. The NICE guideline recommendations relevant to community midwives' practice were used as the basis for the questionnaire development to define midwives' behaviours and SCT constructs. As the guideline recommendations were not written as explicit behaviours, they were adapted to define the Target population, Action, Context and Time (TACT) [29]. Additionally, the adaptations used people-first language to describe the target population as women with obesity rather than obese women [30]. We also incorporated additional behaviours relating to GWG that were not included in the UK guidelines, as the evidence-base suggested health professionals' lack of knowledge about GWG was a barrier to providing weight management support during pregnancy [20]. As UK guidelines did not recommend specific GWG ranges, the behaviours relating to GWG were in the context of discussing GWG risks with women and being able to respond to women's queries about GWG rather than advising women on target GWGs. These adaptations resulted in 32 behaviours, reflecting all six behaviour sub-categories being included in the questionnaire (S2 Table). The behaviours were replicated for the intentions construct, adapting each statement from *"I do the behaviour"* to *"I intend to do the behaviour"* (n = 32 questions). Due to the volume of questions and potential for participant burden, the behaviours selected for the self-efficacy and outcome expectancies constructs were

prioritised, based on the two SCT models, to reflect the key barriers. There were 21 behaviours included in the self-efficacy questions, and 19 for outcome expectancies. The questions relating to outcome expectancies also required the inclusion of a consequence of doing the behaviour: *"if I do X, then Y will happen";* these were informed by the SCT models (e.g. *"if I discuss weight status with pregnant women who have an obese BMI, then it will negatively impact on the relationship I have with them"*). The evidence-base of midwives' perspectives also identified that their own weight status has an impact on discussing weight with pregnant women [31]. We asked midwives to identify how they perceived their own weight (very underweight, underweight, healthy weight, overweight, very overweight) and whether this made it easier or harder to discuss weight status and risks with pregnant women on a 7-point Likert scale (1 = much harder, 7 = much easier).

Health professionals' attitudes were identified as evidence-based determinants of their behaviour. These were primarily related to negative obesity stereotypes and prejudices which are known to be informed, in part, by a lack of knowledge about the causes of obesity [32]. The GLOWING intervention included elements relating to the causes of obesity and the implications of obesity-related stigma in pregnancy. The Beliefs About Obese Persons Scale (BAOP) was included in both the pre- and post-intervention questionnaires. This is a 10-item scale, answered on a 6-point Likert scale, that measures the extent that one believes obesity is under the control of the person living with obesity; it was designed to address the lack of psychometrically adequate measures of beliefs about people living with obesity [33]. The potential score ranges from 0–48. Higher scores indicate that participants believe obesity is largely beyond the individuals' control and therefore they tend to have more positive attitudes toward people living with obesity than those who believe obesity can be controlled by the individual.

## Data analysis

All data entry was carried out in duplicate. Two people entered the data independently, compared the data entered to identify potential data entry errors, and any discrepancies in data entry were validated against the original questionnaires. Descriptive analysis was carried out to compare the data with the theoretical models we had developed by pooling the data for all midwives pre-intervention for the SCT and BAOP questionnaires. A descriptive analysis was also carried out to explore midwives' perception of their own weight status, and whether they felt this made it easier or harder to discuss pregnant women's weight status and obesity risks.

## SCT questionnaires

The data for the behaviour items and SCT constructs were transformed to a 0–100 scale to enable comparisons across all constructs. The sum scores for the 7-point Likert scales were calculated by combining the individual questions within the two behaviour categories and for each SCT construct (S3 Table). The theoretical minimum and range for each sum score was calculated and used to transform each category to the 0–100 scale. For example, if 10 questions were included in the sum score, then the minimum score if all 10 questions scored the lowest value of 1 would be 10, the maximum score if all ten questions scored the highest value of 7 would be 70, giving a theoretical range of scores as being 60. The 0–100 scale conversion was calculated for each behaviour category and SCT construct using the following formula whereby a higher score indicated a more positive outcome (e.g. greater self-efficacy): ((SUM score-theoretical minimum value)/theoretical range) x 100.

The same process was carried out at the behaviour sub-category level to explore whether there were any specific areas of midwifery practice driving the overall category scores. Descriptive statistics were calculated to examine the mean and standard deviation (SD) of the scores at

baseline and follow up, for the intervention and control arms. Mean difference in the change in pre- and post-intervention scores were estimated for midwives who returned both questionnaires, alongside 95% confidence intervals (CI).

Internal consistency of the questionnaire was measured using Cronbach's Alpha, pooling the intervention and control arm data at baseline. The analysis was applied to the data within the communication-related behaviours and support/intervention-related behaviours for each SCT construct to assess whether midwives who scored low or high on one question also scored similarly on other questions within that group. An overall Cronbach's Alpha score of 0.8 or above, a corrected item total correlation of 0.4 or above, and if the Cronbach's Alpha score for any deleted variable was lower than the overall value, was considered to represent good internal consistency [34].

## Research ethics

The study was approved by the Proportionate Review Sub-committee of the Yorkshire & The Humber—South Yorkshire Research Ethics Committee (ref: 15/YH/0565, 16/12/2015). All midwives were provided with an information sheet explaining the study, and written informed consent was provided before completing the questionnaire. For midwives in the intervention arm, an additional information sheet was provided and written informed consent was provided before delivery of the GLOWING intervention. Midwives had a unique participant ID number to maintain anonymity and could request to withdraw from the study using this ID number as their reference.

## Results

### Participants

One hundred midwives were randomly selected to complete the GLOWING questionnaires (n = 49 intervention, n = 51 control arm), 74 consented and 68 (92%) of those who consented returned questionnaires: 33 (87%) midwives in the intervention and 35 (97%) in the control arm. Of these, 47 (69%) returned both the pre- and post-intervention questionnaires (14 intervention and 33 control), 19 (28%) returned only the pre-intervention questionnaire (18 intervention and one control), and two (3%) returned only their post-intervention questionnaire (one intervention and one control). The key reason for loss to follow up in the intervention arm was related to the length of the questionnaires and time required to complete them, on top of attending the GLOWING intervention, completing evaluation form and participating in focus groups. When comparing the characteristics of midwives who returned both questionnaires compared with those who only returned one questionnaire, there was little difference in their age, sex, ethnic group, number of years practicing as a midwife, or their clinical speciality (S4 Table). A higher proportion of the midwives who only returned one questionnaire were working full-time (68.4%) than those who returned both questionnaires (51.1%). There was little difference in the baseline scores for the behaviour categories and SCT constructs between the midwives who returned one or both questionnaires (S4 Table).

The personal characteristics of midwives who completed pre-intervention questionnaires showed that all were female with a mean age of 46 years (SD 8.4), most were white (97%) and perceived their own weight to be slightly or very overweight (57.6%) (Table 1). Overall, the mean length of time practicing as a midwife was 19.6 years (SD 12.2), most were working as community midwives (93.8%), and there was a mix of midwives working full-time and part-time (56.1% and 43.9% respectively). There were some differences between midwives in the control and intervention arms. The intervention arm had a smaller proportion of midwives who perceived their weight as being healthy, and a higher proportion who perceived their

**Table 1. Personal and professional characteristics of midwives completing pre-intervention questionnaires.**

| | Intervention (n = 32) | Control (n = 34) | Total (n = 66) |
|---|---|---|---|
| Age (mean, SD) | 47.6 (6.8) | 45.4 (9.5) | 46.4 (8.4) |
| Gender female (n, %) | 32 (100) | 34 (100) | 66 (100) |
| Ethnic group (n, %) | | | |
| White | 31 (96.9) | 33 (97.1) | 64 (97.0) |
| Other | 0 | 1 (2.9) | 1 (1.5) |
| Prefer not to answer | 1 (3.1) | 0 | 1(1.5) |
| Perception of their own weight | | | |
| Very underweight | 0 | 0 | 0 |
| Slightly underweight | 1 (3.1) | 0 | 1 (1.5) |
| Healthy weight | 11 (34.4) | 16 (47.1) | 27 (40.9) |
| Slightly overweight | 15 (46.9) | 16 (47.1) | 31 (47.0) |
| Very overweight | 5 (15.6) | 2 (5.9) | 7 (10.6) |
| Number years in practice (mean, SD) | 20.9 (13.8) | 18.3 (10.4) | 19.6 (12.2) |
| Employed (n, %) | | | |
| Full-time | 21 (65.6) | 16 (47.1) | 37 (56.1) |
| Part-time | 11 (34.4) | 18 (52.9) | 29 (43.9) |
| Speciality (n, %) | | | |
| Community | 28 (90.3) | 33 (97.1) | 61 (93.8) |
| Other | 3 (9.7) | 1 (2.9) | 4 (6.2) |

SD = standard deviation

weight as very overweight or slightly underweight, who worked full-time and in non-community roles (Table 1).

When asked to consider whether their own weight made it easier, harder, or made no difference to having weight-related communication with pregnant women, the greatest proportion of midwives felt that their weight made it easier to discuss women's weight status (43.1%) and obesity risks (50.0%) (S5 Table). Among the midwives who perceived their weight as being slightly or very overweight, the greatest proportion felt this made it easier to discuss weight status and obesity risks with pregnant women, although this was increased in the intervention arm (70.0% weight status and 73.7% risk communication) compared to the control arm (44.4% for both weight status and risk communication). Among the midwives who perceived they had a healthy weight, the greatest proportion reported that this made no difference to discussing weight status and obesity risk in both the intervention arm (45.5% for both weight status and risk communication) and control arm (43.8% and 50.0% respectively).

## Questionnaire data

The Cronbach's Alpha analysis demonstrated good internal consistency for communication-related behaviours and support/intervention-related behaviours for all SCT constructs (Cronbach's Alpha ranging from 0.82–0.96; S6 Table).

**Comparing pre-intervention data with theoretical models.** Pre-intervention, the midwives' scores for the communication-related behaviours were generally higher overall than the scores for the support/intervention-related behaviours (Table 2). The intention and behaviour constructs for communication-related behaviours were high-scoring pre-intervention. For the support/intervention-related behaviours, the self-efficacy and behaviour constructs were the lowest-scoring constructs overall. These data reflect the evidence-based theoretical models we developed on which the intervention development was based. The models showed that self-

**Table 2. Summary of baseline social cognitive theory data grouped by behaviour category.**

| | | Self-efficacy, mean (SD)[b] | Outcome expectancies, mean (SD)[b] | Intention, mean (SD)[b] | Behaviour, mean (SD)[b] |
|---|---|---|---|---|---|
| **Communication-related behaviours** | Weight | 69.0 (19.6) | 67.1 (16.7) | 89.6 (14.5) | 87.3 (16.0) |
| | Risk | 71.5 (15.8) | 64.9 (15.4) | 83.4 (15.7) | 72.5 (17.9) |
| **Support / intervention-related behaviours** | Diet / nutrition | 53.2 (18.5)[a] | 70.2 (22.2)* | 71.3 (19.6) | 66.6 (19.5) |
| | Physical activity | | | 65.0 (20.6) | 48.1 (21.2) |
| | Weight management | 50.5 (21.7) | 74.1 (20.4) | 61.4 (22.1) | 44.6 (23.5) |
| | Referral / signposting | 46.5 (21.3) | 58.0 (23.3) | 72.3 (20.7) | 62.2 (21.6) |

SD = standard deviation

[a] Due to the limited number of diet and nutrition questions for self-efficacy and outcome expectancies, these were combined with the physical activity questions when creating the scale scores for these constructs.

[b] Scores ranged between 0–100 with a higher score indicating a more positive outcome (e.g. higher self-efficacy).

efficacy and outcome expectancies were the key barriers to communication-related behaviours, and that self-efficacy was the key barrier to support/intervention-related behaviours. The majority of barriers to practice identified in the models related to the support/intervention-related behaviours, which is also reflected in these baseline data which overall were lowest scoring. The pre-intervention mean BAOP scores were low (14.6, SD 5.7) reflecting a tendency towards beliefs that obesity is within the individual's control.

**Comparing intervention and control arms.** The descriptive statistics for pre- and post-intervention data are shown in Table 3, split by the communication- and support/intervention-related categories. The mean scores for the communication-related category were higher overall than for the support/intervention-related category. Communication-related behaviours

**Table 3. Pre- and post-intervention scores for communication- and support/intervention-related behaviours.**

| | Intervention[a] | | Control[a] | | Pre-post intervention difference[b] (MD, 95% CI) |
|---|---|---|---|---|---|
| | Pre-intervention (mean, SD)[c] | Post-intervention (mean, SD)[c] | Pre-intervention (mean, SD)[c] | Post-intervention (mean, SD)[c] | |
| **Communication-related behaviours** | | | | | |
| Self-efficacy | 68.0 (15.9) | 78.1 (16.3) | 72.7 (16.0) | 72.6 (19.4) | 8.94 (-1.02, 18.90) |
| Intention | 87.6 (13.8) | 91.1 (11.8) | 85.5 (14.9) | 86.3 (13.6) | 3.05 (-3.58, 9.69) |
| Outcome expectancies | 67.3 (14.5) | 73.0 (17.0) | 64.3 (15.9) | 63.9 (18.0) | 6.18 (-2.43, 14.78) |
| Behaviours | 81.0 (11.4) | 84.7 (12.9) | 78.9 (17.4) | 78.5 (14.8) | 4.58 (-2.65, 11.82) |
| **Support/intervention-related behaviours** | | | | | |
| Self-efficacy | 48.5 (16.6) | 71.4 (17.1) | 52.2 (21.0) | 58.4 (20.1) | 17.92 (7.78, 28.07) |
| Intention | 65.5 (17.5) | 85.4 (13.7) | 66.5 (19.2) | 73.6 (18.7) | 12.68 (2.76, 22.59) |
| Outcome expectancies | 67.5 (18.2) | 77.0 (17.6) | 71.7 (15.0) | 69.7 (13.2) | 9.85 (-1.06, 20.75) |
| Behaviours | 49.7 (14.5) | 62.0 (15.6) | 54.7 (21.6) | 60.6 (18.7) | 6.08 (-2.56, 14.73) |

SCT = social cognitive theory, SD = standard deviation, MD = mean difference, CI = confidence interval

[a] Mean and SD calculated for all midwives returning any questionnaire

[b] Mean difference and 95% CI calculated for midwives who returned both pre- and post-intervention questionnaires

[c] SCT questionnaire scores ranged from 0–100; BAOP questionnaire scores ranged from 0–48; for both questionnaires a higher score reflects a more positive outcome

and intentions were high in both the intervention and control arms, pre- and post-intervention, suggesting there may be a ceiling effect for these constructs. In the control arm, there was limited difference between the pre- and post-intervention scores for both behaviour categories. Post-intervention, the scores were consistently higher in the intervention arm than the control arm, particularly for support/intervention self-efficacy (mean scores 71.4 (SD 17.1) and 58.4 (SD 20.1) respectively) and intentions (mean scores 85.4 (SD 13.7) and 73.6 (SD 18.7) respectively). When comparing mean difference in scores among midwives returning both pre- and post-intervention questionnaires, a similar pattern was observed with the support/intervention self-efficacy and intentions showing the greatest change (MD 17.92 (95% CI 7.78, 28.07) and 12.68 (95% CI 2.76, 22.59) respectively).

When breaking down the behaviour categories further to explore pre- and post-intervention scores, we observed that intervention midwives' mean self-efficacy was higher post-intervention than pre-intervention for all behaviours (i.e. weight communication, risk communication, diet and nutrition, PA, weight management and referrals and signposting behaviours) and also consistently higher than the control arm post-intervention (S7 Table). Intervention arm midwives' self-efficacy was particularly increased for the guideline recommended diet, nutrition, PA and weight management behaviours, which showed the largest increase in scores pre- to post-intervention (MD 24.77 (95% CI 14.09, 35.44) and 18.88 (95% CI 7.88, 29.88) respectively. Post-intervention outcome expectancies and intentions relating to diet and nutrition, PA, and weight management were also markedly increased in the intervention arm (compared with pre-intervention and control arm scores), but to a lesser extent than self-efficacy.

The mean scores for the BAOP questionnaire were similar pre-intervention in both the intervention (mean 14.1, SD 4.4) and control arms (mean 15.1, SD 6.8), although slightly lower in the intervention arm. The scores were slightly higher in the intervention arm post-intervention (mean 16.1, SD 7.2), whereas there was minimal change in scores in the control arm (mean 15.2, SD 6.6).

## Discussion

This paper reports the descriptive results of the GLOWING pilot trial relating to midwives reporting of their self-efficacy, outcome expectancies, intentions and routine behaviours in the context of UK guidelines for weight management during pregnancy. The data reported in this study provide some proof of concept for the evidence-based theoretical models that were developed to underpin the GLOWING intervention. Prior to GLOWING, the evidence-base suggested that health professionals' low self-efficacy was central to the barriers and facilitators to the implementation of the UK guideline recommendations, and that the majority of barriers were related to the support/intervention-related behaviours. This is reflected in the GLOWING data which demonstrated that, pre-intervention, midwives' self-efficacy was the lowest scoring construct across all behaviours, and the support/intervention-related behaviours tended to score lower than the communication-related behaviours. The GLOWING intervention was developed to address the evidence-based barriers incorporated in the SCT models. There was a particular emphasis on improving midwives' self-efficacy for both communication- and support/intervention-related behaviours, but with more focus on the support/intervention related barriers to practice. The descriptive data reported by midwives in the GLOWING pilot trial suggests that there was limited change in self-efficacy for the control arm, whereas the self-efficacy scores increased in the intervention arm and were consistently higher than the control arm post-intervention. This was apparent for both communication- and support/intervention-related behaviours, but more so for the support/intervention-related

behaviours. The pilot trial was not powered to be able to detect a significant difference between the intervention and control arms at follow-up or change in self-efficacy from pre- to post-intervention. However, the data are suggestive that the intervention may be impacting on the target construct of self-efficacy, particularly for the support/intervention-related behaviours.

The data also suggested a potential ceiling effect for midwives reporting of intention and behaviour constructs for communication-related behaviours. Some of these behaviours (e.g. relating to measuring and discussing BMI at the booking appointment) are now embedded into routine care. The BMI measurement influences the further discussions and referrals required relating to clinical management of pregnancy, such as referral for routine screening for gestational diabetes or consultant obstetrician led care [35]. While the evidence-base suggests that weight communication might be perceived by midwives as a difficult conversation to have (i.e. impacting on self-efficacy and outcome expectancy constructs), they may feel that this is part of their professional role and routine care, which could explain the potential ceiling effect relating to these constructs and behaviours. However, the support/intervention-related behaviours are not necessarily part of routine clinical care pathways, and midwives may feel less conflict reporting lower intention or behaviour relating to these, as demonstrated in the data, due to it not being an explicit part of their professional role. If not part of routine care, then they may have less experience performing these behaviours, influencing their self-efficacy, and negative experiences when performing these behaviours less frequently may have influenced their outcomes expectancies (e.g. women getting upset).

This paper also reports data on midwives' beliefs about people living with obesity using the BAOP scale. There was a similarity in pre-intervention BAOP data from midwives in this study (mean score 14.6 SD 5.7) to published data from a general population of UK adults (mean score 14.7 SD 6.7) [36], suggesting that midwives hold similar views about people living with obesity to the wider society. Although the GLOWING pilot study was not powered to detect a statistically significant change in BAOP scores, there was an increase in scores in the intervention arm and no change in the control arm. However, this change was minimal and unlikely to have any meaningful clinical difference on practice. A recent meta-analysis of 17 studies reported that health professionals hold implicit and/or explicit weight-biased attitudes toward people with obesity [37], highlighting the continued need to address this issue with health professionals. The elements of the GLOWING intervention addressing weight bias could be strengthened in light of this descriptive preliminary data and evidence-base of health professionals' ongoing need for support.

Despite the wealth of evidence of the multiple and complex barriers to practice for health professional guideline implementation relating to maternal obesity and weight management practice [20], an absence of curricula in universities [38], and a call to action for midwifery education and training [39], there is an absence of adequately powered trials of interventions in this field. There have been before and after studies published with no control arm, primarily feasibility studies. In the UK, a feasibility study of a compact midwifery training intervention among 32 practicing midwives resulted in increased knowledge and confidence relating to NICE guideline behaviours [40], and an online programme delivered to 52 final year midwifery students [41] increased students' subjective norms, perceived behavioural control, and knowledge of BCTs *"to discuss lifestyle change with obese patients"*, but not their intention or attitudes. In Australia, an online CPD course with before and after questionnaires completed by 36 health professionals identified increased perception of the importance of weight management for pregnancy and confidence to provide advice, but no difference in knowledge [42]. Whilst these before and after studies report some positive data, there are limitations relating to the lack of a control arm and small sample sizes. Although the pilot GLOWING showed minimal difference between pre- and post-intervention data in the control arm, there were some

small changes (although the study was not powered to know if these were statistically significant). It is possible that the passage of time, and midwives' exposure to events or resources unrelated to the intervention, could result in changes in midwives' practice, confidence, knowledge etc. which need to be accounted for using a control arm, or factored into statistical analysis of single arm studies. A definitive trial with adequate power is required to determine the effectiveness and cost-effectiveness of implementation interventions relating to maternal obesity and weight management.

There are strengths and limitations to this research. The intervention was developed following a rigorous approach using evidence-based theoretical models. A recent scoping review of implementation interventions in the maternity context identified that out of 158 published studies, only 14 reported the use of a theory, model and/or framework, and these typically guided data analysis or data collection rather than the design of the study [43], demonstrating the novelty and rigour of the GLOWING study in comparison to others in this field. The questionnaire used to collect data on the SCT constructs had to be developed to be tailored to this study and therefore was not a validated questionnaire; however, it demonstrated good internal consistency. However, this paper reports the descriptive results of a pilot trial, and therefore it is not powered to determine any statistically significant change in midwives reporting of the SCT constructs or BAOP questionnaire. There was loss to follow up, particularly in the intervention arm, which we believe to be primarily related to intervention fatigue and participant burden. However, there were minimal differences in socio-demographics or baseline measurements of SCT constructs between the midwives lost to follow up and those who returned both questionnaires. This paper reports only quantitative data which does not fully reflect the mechanisms. However, we have also conducted a qualitative process evaluation of midwives' experiences of the intervention, and their views on the impact of the intervention on routine care which will be reported separately. The GLOWING data for pilot trial feasibility will also be reported separately.

## Conclusions

The GLOWING pilot trial data provides proof of concept of the theoretical models used to inform its development. The descriptive data reported in this paper suggests that the intervention may be successfully targeting self-efficacy as it was designed to do. A definitive trial with adequate power is required to determine the effectiveness and cost-effectiveness of the GLOWING intervention.

## Supporting information

**S1 File.**
(DOCX)

**S1 Fig. Flowchart of NICE Guideline behaviours relevant to midwifery practice grouped into thematic behaviour categories and sub-categories for communication-related behaviours and behaviour support/intervention-related behaviours.** Note, Behaviour categories and sub-categories are: 1. Communication-related behaviours (sub-categories: weight communication and risk communication) 2. Support and intervention-related behaviours (sub-categories diet and nutrition, physical activity, weight management, referrals and signposting).
(DOCX)

**S1 Table. Standards for reporting implementation studies: The StaRI checklist.**
(DOC)

**S2 Table. Adapted NICE guideline recommended behaviours developed for the GLOW-ING questionnaire.**
(DOCX)

**S3 Table. Sum score behaviour categories derived from the questionnaire items for each SCT construct.** * Questions required reverse coding.
(DOCX)

**S4 Table. Midwife characteristics and scores for the behaviour categories and Social Cognitive Theory (SCT) constructs, comparing those who returned one or both questionnaires.**
*Note, 21 midwives only returned one questionnaire, two of these only returned their post-intervention questionnaire therefore have missing pre-intervention data.
(DOCX)

**S5 Table. Midwives perceptions of the impact of their own weight on the level of difficulty the experience discussing weight status and risks of obesity with pregnant women.**
HW = healthy weight, OW = slightly overweight/very overweight *HW in the intervention arm includes one midwife who perceived their own weight as slightly underweight.
(DOCX)

**S6 Table. Internal validity of the questionnaire items for each behaviour category and social cognitive theory construct.** *Communication-related behaviours include weight communication and risk communication; support/intervention-related behaviours include diet and nutrition, physical activity, weight management, and referrals and signposting.
(DOCX)

**S7 Table. A comparison of pre- and post-intervention scores for communication- and support/intervention-related behaviour sub-categories and social cognitive theory constructs.**
SD = standard deviation * Due to the limited number of diet and nutrition questions for self-efficacy and outcome expectancies, these were combined with the physical activity questions when creating the sum scores for these constructs.
(DOCX)

## Acknowledgments

We would like to thank Anita Tibbs, Phoebe Orangu and Zoe Bell for contributing towards duplicate data entry and validation. We thank the research midwife teams at the participating NHS Trusts for facilitating the questionnaire distribution to midwives, Judith Ormonde, Gayle Gills, Helen Howlett, Christine Moller-Christensen and Caitlin Barry. We would like to thank Dr Lucia Rehackova for contributing to mapping the evidence-based barriers and facilitators to practice to the SCT models and intervention development.

## Author Contributions

**Conceptualization:** Nicola Heslehurst, Falko F. Sniehotta, Judith Rankin, Elaine McColl.

**Data curation:** Nicola Heslehurst, Catherine McParlin.

**Formal analysis:** Nicola Heslehurst, Catherine McParlin.

**Funding acquisition:** Nicola Heslehurst, Elaine McColl.

**Investigation:** Nicola Heslehurst, Catherine McParlin.

**Methodology:** Nicola Heslehurst, Falko F. Sniehotta, Judith Rankin, Elaine McColl.

**Project administration:** Nicola Heslehurst.

**Supervision:** Falko F. Sniehotta, Judith Rankin, Elaine McColl.

**Validation:** Catherine McParlin.

**Writing – original draft:** Nicola Heslehurst, Catherine McParlin.

**Writing – review & editing:** Nicola Heslehurst, Catherine McParlin, Falko F. Sniehotta, Judith Rankin, Elaine McColl.

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
