## [Decision Letter · Decision Letter 0]

30 Mar 2022

PONE-D-21-39444Midwives’ survey of their weight management practice before and after the GLOWING guideline implementation intervention: a pilot cluster randomised controlled trialPLOS ONE

Dear Dr. Heslehurst,

Thank you for submitting your manuscript to PLOS ONE. After careful consideration, we feel that it has merit but does not fully meet PLOS ONE’s publication criteria as it currently stands. Therefore, we invite you to submit a revised version of the manuscript that addresses the points raised during the review process.

We look forward to receiving your revised manuscript.

Kind regards,

Shahrad Taheri

Academic Editor

PLOS ONE

Journal Requirements:

We will update your Data Availability statement on your behalf to reflect the information you provide."

Reviewers' comments:

Reviewer's Responses to Questions

**Comments to the Author**

1. Is the manuscript technically sound, and do the data support the conclusions?

Reviewer #1: Yes

Reviewer #2: Yes

Reviewer #3: Yes

2. Has the statistical analysis been performed appropriately and rigorously? 

Reviewer #1: Yes

Reviewer #2: I Don't Know

Reviewer #3: Yes

3. Have the authors made all data underlying the findings in their manuscript fully available?

Reviewer #1: No

Reviewer #2: Yes

Reviewer #3: Yes

4. Is the manuscript presented in an intelligible fashion and written in standard English?

Reviewer #1: Yes

Reviewer #2: Yes

Reviewer #3: Yes

5. Review Comments to the Author

Reviewer #1: This is very interesting and an excellent read. I'm particularly pleased that you included the perceived weight status of the midwifes as this is rarely considered. The intervention was informed by psychology behaviour change theory and the questionnaire measures were appropriate.

Perhaps non-parametric tests would have been feasible to run and report on this data, though the descriptive are well presented.

This is a very well written article. I have no requests or recommendations for amendments.

Publish as submitted. Well done.

Reviewer #2: Important note: This review pertains only to ‘statistical aspects’ of the study and so ‘clinical aspects’ [like medical importance, relevance of the study, ‘clinical significance and implication(s)’ of the whole study, etc.] are to be evaluated [should be assessed] separately/independently. Further please note that any ‘statistical review’ is generally done under the assumption that (such) study specific methodological [as well as execution] issues are perfectly taken care of by the investigator(s). This review is not an exception to that and so does not cover clinical aspects {however, seldom comments are made only if those issues are intimately / scientifically related & intermingle with ‘statistical aspects’ of the study}. Agreed that ‘statistical methods’ are used as just tools here, however, they are vital part of methodology [and so should be given due importance].

COMMENTS: What exactly you want to convey when you say [in ‘Abstract-Background’] that “The GLOWING intervention uses social cognitive theory (SCT) to address evidence-based barriers to practice, aiming to support midwives’ implementation of guidelines” and later [in ‘Abstract-Methods’] say “UK guidelines were grouped into communication-related behaviours (weight- and risk-communication) and support/intervention-related behaviours (diet/nutrition, physical activity, weight management, referrals/signposting)” and also say that “randomised to intervention (all eligible midwives received the intervention) or control (no intervention)”. Is not this account confusing? At least for me it is confusing. I could not really be clear about why “UK guidelines were grouped into two” and which is the intervention? Please make me [average reader] understand clearly.

In fact, when the allocation is ‘randomly’ done, statistical comparison of baseline characteristics is not indicated. In this context, please read the following which is pasted from one standard textbook on ‘Research Methodology’.

Statistical comparison of baseline characteristics is not desirable at all [because even if P-value turns out to be significant (while comparing baseline characteristics despite random allocation), it is, by definition, a false positive] as you then are supposed to be testing ‘randomization’ then, which in any single trial may not balance all baseline characteristics because ‘randomization’ is a sort of ‘insurance’ and not a guarantee scheme.

Though the measures/tools used are appropriate [line 237-: SCT questionnaires], most of them yield data that are in [at the most] ‘ordinal’ level of measurement [and not in ratio level of measurement for sure {as the score two times higher does not indicate presence of that parameter/phenomenon as double (for example, a Visual Analogue Scales VAS score or say ‘depression’ score)}]. Then application of suitable non-parametric test(s) is/are indicated/advisable [even if distribution may be ‘Gaussian’ (i.e. normal)]. Agreed that there is/are no non-parametric test(s)/technique(s) available to be used as alternative in all situation(s) [suitable / most desired/applicable], but should be used whenever/wherever they are available.

This being a ‘pilot’ study/trial, ‘sample size’ is not an issue. However, from the account given in lines 43-44 [post-intervention, mean (SD) scores were consistently higher among intervention midwives than controls] it seems that you have compared post-intervention (for both) only. Is that by ANCOVA so that Pre-intervention/baseline scores are taken cognizance of? Even if baselines are similar {not significantly different}, one should take pre scores (baseline) into consideration. Why not work on ‘change score(s)’? According to section ‘Comparing intervention and control arms’ [lines 344-354: The descriptive statistics for pre- and post-intervention data are shown in Table 3, split by the communication- and support/intervention-related categories. The mean scores for the communication-related category were higher overall than for the support/intervention-related category. Communication-related behaviours and intentions were high in both the intervention and control arms, pre- and post-intervention, suggesting there may be a ceiling effect for these constructs. In the control arm, there was limited difference between the pre- and post-intervention scores for both behaviour categories. Post-intervention, the scores were consistently higher in the intervention arm than the control arm, particularly for support/intervention self-efficacy (mean scores 71.4 (SD 17.1) and 58.4 (SD 20.1) respectively) and intentions (mean scores 85.4 (SD 13.7) and 73.6 (SD 18.7) respectively)] you have compared ‘pre- and post-intervention scores’ separately for both groups. Whereas the section heading ‘Comparing intervention and control arms’ implies/indicates between groups comparison.

May please read the following which is again pasted from the same textbook on ‘Research Methodology’.

Many randomized trials involve measuring at baseline and after treatment. In few papers [example: ‘The use of percentage change from baseline as an outcome in a controlled trial is statistically inefficient – a simulation study’ by Andrew Vickers in BMC Medical Research Methodology 2001, 1:6], compared all four possibilities for how such trials can be analyzed: only post–treatment; absolute change between baseline and post-treatment (called as ‘change score’ method); percentage change between baseline and post-treatment (called as ‘percent change score’ method) and analysis of covariance (ANCOVA) with baseline score as a covariate. The statistical power of each method was determined for a hypothetical randomized trial under a range of correlations between baseline and post-treatment scores. ANCOVA has been found to have the highest statistical power.

‘Strengths and limitations’ of this research given in lines 459 onwards are appreciated. Though, as pointed out in ‘important note’ above “This review pertains only to ‘statistical aspects’ of the study and so ‘clinical aspects’ should be assessed separately/independently, in my opinion, this study/article has great(er) potential and to rescue this article (which is quite possible), some amount of re-vision/re-drafting (may be considering above highlighted points) is needed, I guess.

Reviewer #3: Thank you for providing me with the opportunity to review this paper. It is a relevant subject of considerable interest as effective implementation of weight management interventions is highly important. The authors have used a pre-existing theoretical framework for their intervention. The results provide important information about how the level of self-efficacy in midwives may be raised, facilitating implementation of guidelines.

This is a well-written paper and the findings are presented in a clear way. I have added some comments about how some details may be clarified, else I believe the article is ready for publishing.

Introduction

Line 75-77, is a long sentence. Perhaps divide it in two?

Line 86-87, at the end of the sentence about complex barriers, I suggest add one example of such a barrier.

Line 101, what does rehearsal “external” pilot trial mean? Is it a pilot trial before the pilot before the actual study? Please clarify.

Data collection

Line 214-216, a long sentence. Perhaps divide it in two for increased readability?

Data analysis

Line 228, What does it mean that the data entry was carried out in duplicate? Did two persons conduct the same work and then compare it? Please provide some more details.

Results in general

The number of midwives in the intervention arm, who provided both pre- and post-intervention questionnaires, were very low and thus the results may only be used to generate a hypothesis. I believe the authors have presented this with clarity in the way the results are displayed. However, in the conclusion (line 478), given the small number (14) of midwives who provide pre- and post-intervention data, perhaps change the word “is” to “may be” as in “…that the intervention may be successfully targeting self-efficacy…”.

6. PLOS authors have the option to publish the peer review history of their article (what does this mean?). If published, this will include your full peer review and any attached files.

Reviewer #1: **Yes: **Dr Lisa Newson

Reviewer #2: No

Reviewer #3: No

---

## [Author Response · Author response to Decision Letter 0]

19 May 2022

Response to reviewers submitted in the cover letter, also copied here.

Dear Prof Taheri

Thank you for considering our manuscript “Midwives’ survey of their weight management practice before and after the GLOWING guideline implementation intervention: a pilot cluster randomised controlled trial” for publication in PLOS One. We have responded to each of the peer reviewer comments below using bullet points, with our responses in italics, and included highlighted changes in the revised manuscript. Please note, we found it hard to follow some of the points raised by peer reviewer 2 and have tried our best to interpret and address these comments.

We look forward to hearing from you regarding this revised manuscript submission.

Yours sincerely

Dr Nicola Heslehurst (on behalf of all co-authors).

Reviewer #1:

This is very interesting and an excellent read. I'm particularly pleased that you included the perceived weight status of the midwifes as this is rarely considered. The intervention was informed by psychology behaviour change theory and the questionnaire measures were appropriate.

- Thank you for these positive comments about our work.

Perhaps non-parametric tests would have been feasible to run and report on this data, though the descriptive are well presented.

- We have followed methodological guidance for analysis of pilot studies and carried out descriptive analysis only (CONSORT extension on pilot trials http://www.consort-statement.org/extensions/overview/pilotandfeasibility). We have added a statement to the methods (data analysis) to explain this point.

This is a very well written article. I have no requests or recommendations for amendments.

Publish as submitted. Well done.

- Thank you for these positive comments about our work.

Reviewer #2: 

COMMENTS: What exactly you want to convey when you say [in ‘Abstract-Background’] that “The GLOWING intervention uses social cognitive theory (SCT) to address evidence-based barriers to practice, aiming to support midwives’ implementation of guidelines” 

- We used social cognitive theory as the behaviour change theory in the intervention to address the barriers to practice that we identified in the evidence-base. The aim of the intervention was to support midwives in their implementation of guidelines. We have split this sentence into two to clarify within the limited word allowance of the abstract. 

and later [in ‘Abstract-Methods’] say “UK guidelines were grouped into communication-related behaviours (weight- and risk-communication) and support/intervention-related behaviours (diet/nutrition, physical activity, weight management, referrals/signposting)” and also say that “randomised to intervention (all eligible midwives received the intervention) or control (no intervention)”. Is not this account confusing? At least for me it is confusing. I could not really be clear about why “UK guidelines were grouped into two” and which is the intervention? Please make me [average reader] understand clearly.

- There were multiple recommendations in the guidelines relating to health professionals’ routine practice (i.e. their behaviours). We grouped these into six sub-categories which were further grouped into two broader categories: 1. weight-related communication and 2. risk communication (i.e. communication related guideline recommendations) and 3. diet/nutrition, 4. physical activity, 5. weight management and 6. referrals/signposting (i.e. support/intervention related guideline recommendations). They were grouped into these categories based on similarity in the type of behaviour they reflected (i.e. health professionals’ communication with pregnant women or health professionals’ provision of support/intervention for pregnant women).

- Barriers to midwives implementing all guideline recommendations were addressed in the intervention, and the intervention aimed to support midwives in the intervention arm with implementing all guideline recommendations. There was no randomisation based on the groups of guideline recommendations. 

- With the limited word count permitted for the abstract, we have tried to amend the wording to clarify this further. 

In fact, when the allocation is ‘randomly’ done, statistical comparison of baseline characteristics is not indicated. In this context, please read the following which is pasted from one standard textbook on ‘Research Methodology’. Statistical comparison of baseline characteristics is not desirable at all [because even if P-value turns out to be significant (while comparing baseline characteristics despite random allocation), it is, by definition, a false positive] as you then are supposed to be testing ‘randomization’ then, which in any single trial may not balance all baseline characteristics because ‘randomization’ is a sort of ‘insurance’ and not a guarantee scheme.

- The referee is correct in saying that statistical comparison (I.e., the application of inferential statistics) of baseline characteristics across arms in a randomised controlled trial is inappropriate. However, it is conventional, and indeed recommended in CONSORT 2010, to include a tabular comparison of key baseline characteristics for intervention and control groups, as we have done (see item 15 of CONSORT 2010 - “... The study groups should be compared at baseline for important demographic and clinical characteristics so that readers can assess how similar they were. Baseline data are especially valuable for outcomes that can also be measured at the start of the trial...”)

- The analysis of intervention and control arms at baseline is descriptive. We have not applied any inferential statistics and therefore present no p values. Given the small sample size, and the fact this was a cluster pilot trial (therefore the participants responses may also be clustered due to similarity between individuals within the cluster) with descriptive analysis, we feel it is important to present both baseline and follow up data in this context.

- We have added a statement to the data analysis methods relating to only carrying out descriptive analysis and not performing any inferential statistics, following good practice recommendations for pilot studies and in respect of baseline comparisons in any RCT. 

Though the measures/tools used are appropriate [line 237-: SCT questionnaires], most of them yield data that are in [at the most] ‘ordinal’ level of measurement [and not in ratio level of measurement for sure {as the score two times higher does not indicate presence of that parameter/phenomenon as double (for example, a Visual Analogue Scales VAS score or say ‘depression’ score)}]. Then application of suitable non-parametric test(s) is/are indicated/advisable [even if distribution may be ‘Gaussian’ (i.e. normal)]. Agreed that there is/are no non-parametric test(s)/technique(s) available to be used as alternative in all situation(s) [suitable / most desired/applicable], but should be used whenever/wherever they are available.

- We have followed methodological guidance for analysis of pilot studies and carried out descriptive analysis only (CONSORT extension on pilot trials http://www.consort-statement.org/extensions/overview/pilotandfeasibility). We have added a statement to the methods (data analysis) to explain this point.

- Additionally, the approach used in calculating sum scores and then converting to a 0-100 metric to come up with an overall subscale score is extremely common in psychometrics, when the individual items are responded to on an interval scale, and there may be a variable number of items per subscale. For instance, it is the method used to calculate subscale scores for the ubiquitous SF-36 measure of QoL (See for example https://c-path.org/wp-content/uploads/2017/05/2017_session5_scoringfinal.pdf )

This being a ‘pilot’ study/trial, ‘sample size’ is not an issue. However, from the account given in lines 43-44 [post-intervention, mean (SD) scores were consistently higher among intervention midwives than controls] it seems that you have compared post-intervention (for both) only. Is that by ANCOVA so that Pre-intervention/baseline scores are taken cognizance of? Even if baselines are similar {not significantly different}, one should take pre scores (baseline) into consideration. Why not work on ‘change score(s)’? According to section ‘Comparing intervention and control arms’ [lines 344-354: The descriptive statistics for pre- and post-intervention data are shown in Table 3, split by the communication- and support/intervention-related categories. The mean scores for the communication-related category were higher overall than for the support/intervention-related category. Communication-related behaviours and intentions were high in both the intervention and control arms, pre- and post-intervention, suggesting there may be a ceiling effect for these constructs. In the control arm, there was limited difference between the pre- and post-intervention scores for both behaviour categories. Post-intervention, the scores were consistently higher in the intervention arm than the control arm, particularly for support/intervention self-efficacy (mean scores 71.4 (SD 17.1) and 58.4 (SD 20.1) respectively) and intentions (mean scores 85.4 (SD 13.7) and 73.6 (SD 18.7) respectively)] you have compared ‘pre- and post-intervention scores’ separately for both groups. Whereas the section heading ‘Comparing intervention and control arms’ implies/indicates between groups comparison.

- While ANCOVA analysis may be appropriate for a definitive trial, this study is a pilot trial and therefore it would not be appropriate to analyse these data in this way. We have followed methodological guidelines relating to the analysis of pilot trial data, and have provided descriptive statistics for both arms at the two data collection time points. 

- We have added a statement to the data analysis methods relating to only carrying out descriptive analysis and not performing any inferential statistics, following good practice recommendations for pilot studies. 

May please read the following which is again pasted from the same textbook on ‘Research Methodology’. Many randomized trials involve measuring at baseline and after treatment. In few papers [example: ‘The use of percentage change from baseline as an outcome in a controlled trial is statistically inefficient – a simulation study’ by Andrew Vickers in BMC Medical Research Methodology 2001, 1:6], compared all four possibilities for how such trials can be analyzed: only post–treatment; absolute change between baseline and post-treatment (called as ‘change score’ method); percentage change between baseline and post-treatment (called as ‘percent change score’ method) and analysis of covariance (ANCOVA) with baseline score as a covariate. The statistical power of each method was determined for a hypothetical randomized trial under a range of correlations between baseline and post-treatment scores. ANCOVA has been found to have the highest statistical power.

- As this is a pilot trial, we have followed methodological guidelines and only carried out descriptive statistics and have not carried out inferential statistics. This would not be appropriate given that the pilot trial is not powered to detect a significant difference between arms. 

- We have added a statement to the data analysis methods relating to only carrying out descriptive analysis and not performing any inferential statistics, following good practice recommendations for pilot studies. 

‘Strengths and limitations’ of this research given in lines 459 onwards are appreciated. Though, as pointed out in ‘important note’ above “This review pertains only to ‘statistical aspects’ of the study and so ‘clinical aspects’ should be assessed separately/independently, in my opinion, this study/article has great(er) potential and to rescue this article (which is quite possible), some amount of re-vision/re-drafting (may be considering above highlighted points) is needed, I guess.

- Thank you for this comment, we gave a great deal of consideration to the discussion of the strengths and limitations of this study to achieve balance, as it is a descriptive analysis of a pilot trial and we need to be careful not to overstate, or attempt to over-analyse the data using inferential statistics which would not be appropriate.

Reviewer #3:

Thank you for providing me with the opportunity to review this paper. It is a relevant subject of considerable interest as effective implementation of weight management interventions is highly important. The authors have used a pre-existing theoretical framework for their intervention. The results provide important information about how the level of self-efficacy in midwives may be raised, facilitating implementation of guidelines. This is a well-written paper and the findings are presented in a clear way. I have added some comments about how some details may be clarified, else I believe the article is ready for publishing.

- Thank you for these positive comments.

Introduction

Line 75-77, is a long sentence. Perhaps divide it in two?

- This sentence has now be divided.

Line 86-87, at the end of the sentence about complex barriers, I suggest add one example of such a barrier.

- An example of the complex barriers has been included.

Line 101, what does rehearsal “external” pilot trial mean? Is it a pilot trial before the pilot before the actual study? Please clarify.

- External rehearsal pilot trials are delivered exactly as they would be if this were a large-scale definitive trial. Additional information has been added to clarify this. 

Data collection

Line 214-216, a long sentence. Perhaps divide it in two for increased readability?

- We have amended this in the manuscript.

Data analysis

Line 228, What does it mean that the data entry was carried out in duplicate? Did two persons conduct the same work and then compare it? Please provide some more details.

- Yes, this means that two people entered the data from the questionnaires into the database independently and their data entry were compared as part of the validation process to check for data entry errors. We have added some additional information to the methods. 

Results in general

The number of midwives in the intervention arm, who provided both pre- and post-intervention questionnaires, were very low and thus the results may only be used to generate a hypothesis. I believe the authors have presented this with clarity in the way the results are displayed. However, in the conclusion (line 478), given the small number (14) of midwives who provide pre- and post-intervention data, perhaps change the word “is” to “may be” as in “…that the intervention may be successfully targeting self-efficacy…”.

- We agree and have changed the wording in the conclusion to “may be” rather than “is”.

---

## [Decision Letter · Decision Letter 1]

1 Sep 2022

PONE-D-21-39444R1Midwives’ survey of their weight management practice before and after the GLOWING guideline implementation intervention: a pilot cluster randomised controlled trialPLOS ONE

Dear Dr. Heslehurst,

Thank you for submitting your manuscript to PLOS ONE. After careful consideration, we feel that it has merit but does not fully meet PLOS ONE’s publication criteria as it currently stands. Therefore, we invite you to submit a revised version of the manuscript that addresses the points raised during the review process.

 ==============================Reviewer #2 still has concerns about the analyses provided in this manuscript. In your response to the reviewers you drew attention to the guidelines provided by CONSORT for pilot data, and argued that you should limit your analyses to descriptive statistics. Reviewer #2 argues that the guidelines do not prevent you from conducting any inferential analyses (although you both agree that the points made about hypothesis-testing for efficacy are clear).

It is certainly the case that the use of term "inferential" in the CONSORT guidelines is open to interpretation.  I am inclined to side with Reviewer 2's reading of the CONSORT extension, and interpret "inferential statistics" narrowly as being about *hypothesis testing* about the effectiveness of an intervention. 

I think, therefore, that the reviewer's requests are consistent with the CONSORT extension statement. To be clear, we do not expect hypothesis testing or estimate of effects with p-value, but an estimate of effect with confidence intervals is appropriate.

The CONSORT statement (e.g.,  item 12a, example 2) discusses CIs: "Typically, any estimates of effect using participant outcomes as they are likely to be measured in the future definitive RCT would be reported as estimates with 95% confidence intervals without P values—because pilot trials are not powered for testing hypotheses about effectiveness."

Therefore, in your resubmission, could you reconsider the requests to conduct some comparisons of the pre/post intervention data?

We look forward to receiving your revised manuscript.

Kind regards,

Steve Zimmerman, PhD

Associate Editor, PLOS ONE

Reviewers' comments:

Reviewer's Responses to Questions

**Comments to the Author**

1. If the authors have adequately addressed your comments raised in a previous round of review and you feel that this manuscript is now acceptable for publication, you may indicate that here to bypass the “Comments to the Author” section, enter your conflict of interest statement in the “Confidential to Editor” section, and submit your "Accept" recommendation.

Reviewer #2: (No Response)

Reviewer #3: All comments have been addressed

2. Is the manuscript technically sound, and do the data support the conclusions?

Reviewer #2: (No Response)

Reviewer #3: Yes

3. Has the statistical analysis been performed appropriately and rigorously? 

Reviewer #2: (No Response)

Reviewer #3: Yes

4. Have the authors made all data underlying the findings in their manuscript fully available?

Reviewer #2: (No Response)

Reviewer #3: Yes

5. Is the manuscript presented in an intelligible fashion and written in standard English?

Reviewer #2: (No Response)

Reviewer #3: Yes

6. Review Comments to the Author

Reviewer #2: COMMENTS: Though few comments were attended, I am not very much convinced for reasons given. According to CONSORT 2010 list available with me & even on NET/WWW “Baseline demographic and clinical characteristics of each group” only. Therefore, no statistical comparison. Whereas it is by you that “However, it is conventional, and indeed recommended in CONSORT 2010, to include a tabular comparison of key baseline characteristics for intervention and control groups, as we have done (see item 15 of CONSORT 2010“. Although, later you said “The analysis of intervention and control arms at baseline is descriptive. We have not applied any inferential statistics and therefore present no p values”, such argument is not desirable/palatable. According to document I have [CONSORT for Pilot trial] though it states that “Formal hypothesis testing for effectiveness (or efficacy) is not recommended. The aim of a pilot trial is not to assess effectiveness (or efficacy) and it will usually be underpowered to do this” it also says “Assessments or measurements to address each pilot trial objective should be the focus of data collection and analysis. This might include outcome measures likely to be used in the definitive trial but, equally, it might not”. True that a pilot trial is not to assess effectiveness (or efficacy) and therefore, no formal hypothesis testing for effectiveness (or efficacy), these guidelines do not prevent you from performing any analyses. In short, I not very convince about “We have followed methodological guidance for analysis of pilot studies and carried out descriptive analysis only”.

And so, I feel, ‘let the respected editor decide the future course’.

Reviewer #3: (No Response)

7. PLOS authors have the option to publish the peer review history of their article (what does this mean?). If published, this will include your full peer review and any attached files.

Reviewer #2: No

Reviewer #3: No

---

## [Author Response · Author response to Decision Letter 1]

26 Sep 2022

Response to peer review and editor comments is provided in the cover letter - copied below:

Dear Prof Zimmerman

Thank you for considering our manuscript “Midwives’ survey of their weight management practice before and after the GLOWING guideline implementation intervention: a pilot cluster randomised controlled trial” for publication in PLOS One. We have responded to the peer reviewer and editors comments below using bullet points, with our response in italics, and included tracked changes in the revised manuscript. 

We look forward to hearing from you regarding this revised manuscript submission.

Yours sincerely

Dr Nicola Heslehurst (on behalf of all co-authors).

Reviewer comment:

Reviewer #2: COMMENTS: Though few comments were attended, I am not very much convinced for reasons given. According to CONSORT 2010 list available with me & even on NET/WWW “Baseline demographic and clinical characteristics of each group” only. Therefore, no statistical comparison. Whereas it is by you that “However, it is conventional, and indeed recommended in CONSORT 2010, to include a tabular comparison of key baseline characteristics for intervention and control groups, as we have done (see item 15 of CONSORT 2010“. Although, later you said “The analysis of intervention and control arms at baseline is descriptive. We have not applied any inferential statistics and therefore present no p values”, such argument is not desirable/palatable. According to document I have [CONSORT for Pilot trial] though it states that “Formal hypothesis testing for effectiveness (or efficacy) is not recommended. The aim of a pilot trial is not to assess effectiveness (or efficacy) and it will usually be underpowered to do this” it also says “Assessments or measurements to address each pilot trial objective should be the focus of data collection and analysis. This might include outcome measures likely to be used in the definitive trial but, equally, it might not”. True that a pilot trial is not to assess effectiveness (or efficacy) and therefore, no formal hypothesis testing for effectiveness (or efficacy), these guidelines do not prevent you from performing any analyses. In short, I not very convince about “We have followed methodological guidance for analysis of pilot studies and carried out descriptive analysis only”.

And so, I feel, ‘let the respected editor decide the future course’.

Editors comment:

Reviewer #2 still has concerns about the analyses provided in this manuscript.

 In your response to the reviewers you drew attention to the guidelines provided by CONSORT for pilot data, and argued that you should limit your analyses to descriptive statistics.

 Reviewer #2 argues that the guidelines do not prevent you from conducting any inferential analyses (although you both agree that the points made about hypothesis-testing for efficacy are clear).

It is certainly the case that the use of term "inferential" in the CONSORT guidelines is open to interpretation. I am inclined to side with Reviewer 2's reading of the CONSORT extension, and interpret "inferential statistics" narrowly as being about *hypothesis testing* about the effectiveness of an intervention. 

I think, therefore, that the reviewer's requests are consistent with the CONSORT extension statement. To be clear, we do not expect hypothesis testing or estimate of effects with p-value, but an estimate of effect with confidence intervals is appropriate.

The CONSORT statement (e.g., item 12a, example 2) discusses CIs:

 "Typically, any estimates of effect using participant outcomes as they are likely to be measured in the future definitive RCT would be reported as estimates with 95% confidence intervals without P values—because pilot trials are not powered for testing hypotheses about effectiveness."

Therefore, in your resubmission, could you reconsider the requests to conduct some comparisons of the pre/post intervention data?

• Based on both peer reviewer and editor feedback, we have amended the manuscript (abstract, methods and results) and relevant supplementary table to include pre and post intervention comparisons, presenting mean differences and 95% confidence intervals

---

## [Decision Letter · Decision Letter 2]

5 Jan 2023

Midwives’ survey of their weight management practice before and after the GLOWING guideline implementation intervention: a pilot cluster randomised controlled trial

PONE-D-21-39444R2

Dear Dr. Heslehurst,

We’re pleased to inform you that your manuscript has been judged scientifically suitable for publication and will be formally accepted for publication once it meets all outstanding technical requirements.

Kind regards,

George Vousden

Staff Editor

PLOS ONE

Additional Editor Comments (optional):

Reviewers' comments:

Reviewer's Responses to Questions

**Comments to the Author**

1. If the authors have adequately addressed your comments raised in a previous round of review and you feel that this manuscript is now acceptable for publication, you may indicate that here to bypass the “Comments to the Author” section, enter your conflict of interest statement in the “Confidential to Editor” section, and submit your "Accept" recommendation.

Reviewer #2: All comments have been addressed

Reviewer #3: All comments have been addressed

2. Is the manuscript technically sound, and do the data support the conclusions?

Reviewer #2: (No Response)

Reviewer #3: (No Response)

3. Has the statistical analysis been performed appropriately and rigorously? 

Reviewer #2: (No Response)

Reviewer #3: (No Response)

4. Have the authors made all data underlying the findings in their manuscript fully available?

Reviewer #2: (No Response)

Reviewer #3: (No Response)

5. Is the manuscript presented in an intelligible fashion and written in standard English?

Reviewer #2: (No Response)

Reviewer #3: (No Response)

6. Review Comments to the Author

Reviewer #2: COMMENTS: Since most important comment made on earlier draft [I, additionally, highly appreciate editor’s nice explanation (excellent job) and support], I recommend the acceptance because now the manuscript has definitely much improved and achieved acceptable level.

Reviewer #3: (No Response)

7. PLOS authors have the option to publish the peer review history of their article (what does this mean?). If published, this will include your full peer review and any attached files.

Reviewer #2: **Yes: **Dr. Sanjeev Sarmukaddam

Reviewer #3: No

---

## [Editor Report · Acceptance letter]

10 Jan 2023

PONE-D-21-39444R2 

Midwives’ survey of their weight management practice before and after the GLOWING guideline implementation intervention: a pilot cluster randomised controlled trial 

Dear Dr. Heslehurst:

I'm pleased to inform you that your manuscript has been deemed suitable for publication in PLOS ONE. Congratulations! Your manuscript is now with our production department. 

Kind regards, 

on behalf of

Dr. George Vousden 

Staff Editor

PLOS ONE